# Responses to Airborne Ozone and Soilborne Metal Pollution in Afforestation Plants with Different Life Forms

**DOI:** 10.3390/plants12163011

**Published:** 2023-08-21

**Authors:** Madeleine S. Günthardt-Goerg, Rodolphe Schläpfer, Pierre Vollenweider

**Affiliations:** 1Swiss Federal Institute for Forest, Snow and Landscape Research WSL, Zürcherstrasse 111, CH-8903 Birmensdorf, Switzerland; pierre.vollenweider@wsl.ch; 2EPFL ENAC IIE Plant Ecology Research Laboratory, GR B2 407 Station 2, CH-1015 Lausanne, Switzerland; rodolphe.schlaepfer@epfl.ch

**Keywords:** biomass, heavy metals, leaf gas exchange, leaf functional traits, light and transmission electron microscopy, macronutrients, ozone, multiple stress, tannins, visible leaf symptoms

## Abstract

**Highlights:**

**What are the main findings?**
Ozone and metal stress caused injuries that were partly similar but differed in their tissue and cell location irrespective of the species.Combined ozone and metal stress showed few interactions.

**What is the implication of the main finding?**
The conifer efficiently blocked the metals in its roots and was more tolerant of ozone stress, resulting in a biomass reduction rating in response to ozone and metal stress: conifer < ruderal forb < deciduous tree.With the current increases in environmental stress, our findings outline the relevance of “slow-return” species strategies, with low productivity but enhanced stress tolerance.

**Abstract:**

With the current increases in environmental stress, understanding species-specific responses to multiple stress agents is needed. This science is especially important for managing ecosystems that are already confronted with considerable pollution. In this study, responses to ozone (O3, ambient daily course values + 20 ppb) and mixed metal contamination in soils (MC, cadmium/copper/lead/zinc = 25/1100/2500/1600 mg kg^−1^), separately and in combination, were evaluated for three plant species (*Picea abies*, *Acer pseudoplatanus*, *Tanacetum vulgare*) with different life forms and ecological strategies. The two treatments elicited similar stress reactions, as shown by leaf functional traits, gas exchange, tannin, and nutrient markers, irrespective of the plant species and life form, whereas the reactions to the treatments differed in magnitude. Visible and microscopic injuries at the organ or cell level appeared along the penetration route of ozone and metal contamination. At the whole plant level, the MC treatment caused more severe injuries than the O3 treatment and few interactions were observed between the two stress factors. *Picea* trees, with a slow-return strategy, showed the highest stress tolerance in apparent relation to an enhancement of conservative traits and an exclusion of stress agents. The ruderal and more acquisitive *Tanacetum* forbs translocated large amounts of contaminants above ground, which may be of concern in a phytostabilisation context. The deciduous *Acer* trees—also with an acquisitive strategy—were most sensitive to both stress factors. Hence, species with slow-return strategies may be of particular interest for managing metal-polluted sites in the current context of multiple stressors and for safely confining soil contaminants below ground.

## 1. Introduction

Tropospheric ozone concentrations currently exceed critical limits in vast regions of the world, posing a major threat to human health, food crop production, and the natural environment [1,2,3,4,5]. Between 2010 and 2014, the highest mean ozone values were measured in the southern USA, Mediterranean Basin, northern India, northwestern and eastern China, Republic of Korea, and Japan [6]. Densely populated regions in emerging countries have become major ozone pollution hotspots [7,8,9,10,11]. Abatements in emissions of nitrogen oxides (NO_x_) and volatile organic compound (VOC) precursors in Europe, North America, and Japan since the end of the last century have resulted in a reduction of ozone peaks, but they have been unable to curb an increase in background ozone concentrations and plant uptake [12,13,14,15]. Global tropospheric ozone concentrations plummeted in response to the worldwide lockdowns and the resulting slowdown of industrial activities during the recent COVID-19 crisis [16], but this improvement was discouragingly short-lived. High levels of air pollution, including ozone, can be accompanied by alarming levels of soil contamination with persistent heavy metals, such as zinc (Zn), lead (Pb), cadmium (Cd), and copper (Cu), especially in populated regions with mining and metal-processing industries [17,18]. Accordingly, European pollution maps indicate the coincidence of serious airborne ozone pollution and soilborne metal contamination throughout industrialised regions [6,19].

Ozone penetrates the intercellular space inside of leaves through stomata, at which point it immediately degrades and thereby triggers the formation of oxidative bursts [20] and the accumulation of reactive oxygen species (ROS) within the apoplast and symplast [21,22,23,24,25]. These ROS cause cell injuries, primarily in assimilative cells, decreasing gas exchange and accelerating cell senescence processes (ACS response; [26,27,28]). They also trigger programmed cell death (PCD) in discrete groups of assimilative cells, resulting in the development of hypersensitive-like reactions (HR-like), similar to responses elicited during some plant–pathogen interactions [29,30]. The severity and dynamics of plant responses depend on the environmental conditions and the synergy with photo-oxidative stress [28,31].

By contrast, interactions between plants and trace metal pollutants begin in roots, where the contaminants can be filtered out (primarily Pb and Cu) or taken up (primarily Cd and Zn) by the root endodermis and then translocated to aboveground organs, ultimately accumulating within foliage [32,33]. Mobile metal contaminants permeate the leaf tissues through the veinlet network and accumulate along the apoplastic and symplastic water and nutrient routes to the water evaporation sites [34] unless they are allocated to safe tissue compartments away from sensitive assimilation and translocation sites [35]. The accumulation of metals such as Cd and Zn causes oxidative stress, activates degenerative processes, and leads to the development of necrosis within foliage, primarily within the lower leaf blade tissues and independent of photo-oxidative stress [36,37]. Morphological and structural indications of HR-like processes have also been reported [27,38], although a full characterisation is still missing and the susceptibility of plant cells to PCD through the accumulation of heavy metals remains unknown. Notwithstanding morphological similarities, the distribution of visible symptoms induced at twig and leaf level by mobile heavy metals show clear differences with those observed in the case of ozone stress: with metal stress, the necrotic dots show spatial connectivity with veins whilst those induced by ozone stress remain strictly intercostal [39,40]. Hence, ozone and heavy metal stress show a mix of contrasts and similarities regarding the affected plant tissues, triggered plant responses, and morphology of injuries. They both cause a reduction in gas exchange and plant growth [37,41,42,43,44] and affect carbon allocation, causing reduced root to shoot ratios [45,46]. However, it remains unclear how plants react to elevated levels of airborne ozone and soilborne heavy metals in combination, as found at brownfield sites in populated regions. Antagonism or additivity of effects represents an important research gap, considering the potential consequences on the phytoremediation performance of spontaneously emerging or planted afforestations at brownfield sites [47].

Beside the specific effects within foliage caused by both stressors, the accumulation of heavy metals at contaminated sites especially affects the nutritional status of reclaiming plants. Plant roots can absorb essential (Cu, Fe, Mo, Mn, Zn) and non-essential (As, Cd, Co, Cr, Ni, Pb [48]) trace metals actively or passively, depending on the availability of each element and irrespective of the plant’s needs [48,49]. Zn is assimilated as hydrated Zn^2+^, organic chelates, or an Fe/Mn oxide complex at the expense of Fe, Mn, and Mg when Zn is in excess [50]. Cu uptake is active in non-polluted environments but becomes passive in toxic conditions, especially if soluble chelates are promoted by low soil pH. Cd absorption involves metabolically passive and active processes, the latter being increased at soil pH < 6. Pb is absorbed by roots but is generally not translocated further into the aboveground plant parts, especially at higher soil pH. By contrast, the absorption and translocation of macronutrients (N, P, K, Ca, Mg) remain little affected by soil contamination. Changes in the nutritional status of foliage that do occur in response to mixed metal contamination can relate to physiological effects of accumulated contaminants and the acceleration of leaf senescence [51]. With ozone stress, only marginal changes in the nutritional status of foliage have been observed [45]. However, early foliage abscission triggered by ozone stress [52] may affect the nutrient pools, similar to other factors enhancing accelerated cell senescence (ACS) [53]. Hence, current evidence from mechanistic experiments considering each stress factor separately suggests that dystrophy effects are practically limited to micronutrients only and primarily result from metal contamination. Given the potential antagonistic/additive effects of combined metal contamination and ozone stress, the consequences of these two environmental stressors on the nutritional status of plants need further research, also considering the wood compartments.

In this study, we used a multifactorial experimental design to compare the responses to different environmental stressors of species representative of the main plant life forms that spontaneously reclaim typical brownfield sites in Central Europe [33]. Our main objective was to differentiate responses to concurrent airborne ozone and soilborne metal pollution from a mechanistic point of view and to relate the observed effects to plant–environment interactions and stress penetration routes within plant organs.

We tested the following hypotheses:

(1) Given their specific penetration routes, ozone and metal contaminants elicit spatially distinct responses in different plant organs (Hypothesis 1a), and their effects are thus additive rather than antagonistic (Hypothesis 1b).

(2) Each stress agent elicits similar types of responses, irrespective of species or plant life form (Hypothesis 2a), but stress severity varies among species (Hypothesis 2b) and plant compartments (Hypothesis 2c).

Therefore, we exposed young trees of two megaphanerophytes and herbaceous plants from one ruderal hemicryptophyte to airborne ozone and soilborne metal contamination—alone or in combination—for one vegetation season within growth chambers, replicating the environmental conditions at a Swiss ozone hotspot. We characterised leaf symptoms, as well as morpho-anatomical, physiological, biochemical, elemental, and biomass responses to the treatments in the root, wood, and foliage organ fractions and related these responses to the penetration mode of the stress agents.

In addition, we investigated the carry-over effects of ozone treatment during the following second year of the experiment, at which point the plant material was exposed to ambient climatic conditions, but the metal treatment was still in effect.

## 2. Results

### 2.1. Visible Injury in Foliage

Characteristic visible injury was observed in response to the treatments in the foliage of all tested species, whilst control plants remained asymptomatic (Figure 1). In *Picea*, the O3 treatment triggered a yellowish-green needle discolouration, first detected 32 days after flushing (AOT40 = 9 ppm·h), which was expressed most on the upper light-exposed side of twigs and more in the current-year (cy) than preceding-year (py) needle generation (Figure 1B and Figure 2A, Table A2). In the MC treatment, both needle generations remained asymptomatic (Figure 1C), whereas ozone injury appeared after a delay of 45 days (AOT40 = 11 ppm·h) in the O3MC trees (significant O3 × MC interaction for cy leaf colour; Figure 1D and Figure 2A, Table A2). The py needles underwent enhanced premature senescence and shedding in response to all stress treatments (Figure 2A and Table A2). The metal contamination had a prevailing effect on py needle shedding in the O3MC treatment, as indicated by similar shedding levels with the MC treatment (significant O3 × MC interaction; Table A2). At the end of the first year of the experiment, this response had caused 9%/17%/17% mass loss in the O3/MC/O3MC treatments. In *Acer*, typical ozone symptoms—in the form of intercostal and adaxial light green dots increasing in severity basipetally—were first detected 64 days (AOT = 13 ppm·h) after flushing in the O3 treatment (Figure 1F). However, this stippling did not significantly change the basic leaf colour at the whole canopy level (Figure 2F, Table A2). The MC treatment caused marked leaf yellowing, originating at the leaf base and progressing along the veins, together with slight downward curling of the leaves (Figure 1G). These symptoms were first detected 45 days after flushing, thus contrasted with the leaf injury response observed in Picea (Figure 1G vs. Figure 1C, Figure 2F vs. Figure 2A, Table A2). Both stippling and vein-linked yellowing were observed in the O3MC treatment, and *Acer* showed more severe injury than *Picea* (Figure 1H vs. Figure 1D). Unlike in *Picea*, only the O3MC treatment tended to cause an acceleration of leaf shedding (Figure 2F). In *Tanacetum*, the O3 treatment accelerated leaf senescence, causing yellow-brown intercostal discolouration (Figure 1J), which first appeared 45 days after flushing. In the MC treatment, the foliage turned yellowish-brown, with conspicuous reddening, upward curling, and desiccation of leaf margins with increasing injury severity (Figure 1K). Injury by both stress factors was attenuated in the O3MC treatment (Figure 1L). Irrespective of the treatment, all *Tanacetum* plants retained their foliage, whilst flowering was reduced (O3) or even missing (MC and O3MC, Figure 1J–L vs. Figure 1I).

During the second year of the experiment, leaf injury in response to the treatments in the overwintered *Picea* and *Acer* trees was similar to that in the first year regarding the symptom morphology. However, it was less severe overall, corresponding to the lower ozone dose (AOT40 = 16 vs. 35 ppm·h). In *Picea*, significant py needle discolouration in trees from the O3 treatment suggested injury aggravation through carry-over effects (cy needle discolouration remained non-significant). In contrast to the preceding year, the steady MC treatment caused yellowing of cy needle tips. In *Acer*, ambient ozone caused similar but milder injury, whereas the injury severity under metal exposure was less severe, reflecting stunted growth and blocked metal uptake.

### 2.2. Structural Injury within Foliar and Root Tissues

Visible injury was underlain by structural changes at the tissue level, which varied according to the stress sensitivity of the affected cells and the penetration route of the stress agent. In direct relation to the visible injury, the foliage of treated plants showed various changes in chlorophyll, polyphenolics, and condensed tannins (Figure 2 and Figure 3, Table A2). In *Picea* mesophyll, the O3 treatment reduced the apparent chlorophyll content in chloroplasts while increasing the vacuolar concentration of polyphenols (Figure 2D and Figure 3B vs. Figure 3A). By contrast, the main effect of the MC treatment was enhanced cell wall lignification in the mesophyll, which was also observed in epi- and hypodermis tissue (Figure 3C vs. Figure 3A), whereas the effects of the two stress factors in the O3MC treatment appeared to be additive (Figure 3D vs. Figure 3B,C). Vacuolar condensed tannins were observed inside of most needle tissues, with little change between treatments (Figure 3E–H). At the organ level, the OPC, PPC, and PPCcw fractions increased by a factor of 4.8, 2.0, and 1.2, respectively, in response to the O3 treatment, whereas the metal exposure had only minor effects on the condensed tannins of needles (Figure 2D, Table A2), consistent with the histochemical observations (Figure 3B–D,F–H). In *Acer* leaf blades, patchy reductions in the chlorophyll content of chloroplasts and increases in the concentration of vacuolar PC were observed in response to both O3 and MC, whereas cell wall lignification was unchanged (Figure 3J–L vs. Figure 3I). With O3 stress, these reactions were primarily observed in the upper mesophyll, rather than in lower leaf blade tissues, as was the case with metal contamination. Vacuolar OPC accumulation in response to the treatments was observed at the same tissue locations as the chlorophyll reductions (Figure 3N,R–T vs. Figure 3J–L). Additionally, large OPC accumulations were found in leaf vein tissues under metal stress (Figure 3O,P). Whilst the O3 treatment slightly reduced the foliar OPC and PPC amounts, the heavy metal exposure caused large increases in OPC/PPC/PPCcw concentrations (by a factor of 8.9/11.2/1.7), effects that were not significantly antagonised by additional ozone stress in the O3MC treatment (no significant O3 × MC interaction; Figure 2I, Table A2). With a significant increase (*Acer*, O3 treatment) or a non-significant decrease (*Picea*, all treatments) vs. the CO treatment, the phenolic oxidation intensity showed a negative correlation with the measured PC concentrations (Figure 2E,J; Table A2). This may explain the inconspicuous contribution of PC accumulation to visible injury in the two treated species (no bronzing symptom; Figure 1).

At the cell ultrastructure level, various degenerative changes were observed within the mesophyll (O3 treatment), as well as the vascular tissues (MC, O3MC treatment), of foliar organs, with minor differences only between the two tree species (Figure 4). In the mesophyll, the cytoplasm showed enhanced condensation, intense vesiculation, and debris leakage into the intercellular space (Figure 4B,D). In chloroplasts, thylakoid and grana membranes showed poor resolution and occasional swelling, whilst enhanced tannin inlays were observed within vacuoles (Figure 4C,D,K vs. Figure 4A,I). In veins, likely phloem necrosis was indicated by cell plasmolysis, cytorrhysis, and debris leakage into the intercellular space, in association with a marked condensation of cell content in both species (Figure 4G,H,O,P vs. Figure 4E, M). In *Acer*, chloroplasts additionally showed a clear increase in the size and density of plastoglobuli (Figure 4K,L), and the upper and lower epidermal cell walls showed increased inlays of dark, presumably phenolic compounds (insets in Figure 4J,K,L vs. Figure 4I). Disruptive changes indicative of HR-like reactions were observed within the upper mesophyll of *Acer* under O3 stress.

In fine roots, degenerative changes were observed within the cortex and central cylinder cells, but in response to soilborne metal contamination only (MC and O3MC treatment). These responses were generally similar to those observed in leaves, regarding both the ultrastructural injury and its severity (Figure A2C,D,G,H,K,L,O,P,S,T,W,X vs. Figure A2A,B,E,F,I,J,M,N,Q,R,U,V). The highest constitutive concentrations of OPC and PPC were measured in fine roots of *Picea* (Figure 2D; root OPC and PPC 20 times and 8 times that in CO needles). The fine-root PC were primarily responsive to metal contamination, with OPC and PPC concentrations 2.3 and 1.5 times the levels in the CO treatment (Figure 2D, Table A2). The much smaller root PPC_cw_ fraction was sizably reduced (−44%/55%/52%) in response to the O3/MC/O3MC treatments. The phenolic oxidation intensity within roots was also lower (−22%/36%/37%) under the O3/MC/O3MC treatments and showed a negative correlation with the OPC and PPC response trend, as observed in foliage (Figure 2D,E, Table A2).

### 2.3. Morphological Changes within Foliage

Given that the treatments began two months before the flushing of new leaves, the morphological development of new cy foliage was potentially responsive to the treatments. However, the LDMC of cy *Picea* needles was unaffected, and only the ozone exposure caused a small increase in LMA (Figure 2B,C, Table A2). By contrast, the LDMC of *Acer* leaves increased significantly in response to the MC treatment (Figure 2G, Table A2). Further, the LMA of *Acer* in the O3 and MC treatments dropped to 31% and 33% of the levels observed in the CO treatment, with a significant antagonistic interaction between O3 and MC resulting in a smaller drop in LMA in the O3MC treatment (Figure 2H, Table A2).

### 2.4. Physiological Effects on Leaf Gas Exchange

Alone or in combination, the ozone exposure and soilborne metal contamination caused significant reductions in the mean daily course of foliar gas exchange. In *Picea*, the surface-based assimilation rates of needles started 1.5 h before those measured in *Acer*, peaked in the morning before maximum photosynthetic photon flux density radiation (PPFD), and decreased progressively afterwards, finally halting 1.5 h after dusk, irrespective of the treatment (Figure 5A and Figure A1C). The mean daily course of transpiration showed a squarer and more symmetric wave, ending one hour later than assimilation (Figure 5B). The ozone and metal exposure reduced transpiration more than net assimilation; consequently, instantaneous needle water use efficiency (WUE) during the daytime (09.00 to 19.00) increased in *Picea* by +50% and +21% in the O3 and MC treatments, respectively, relative to the CO treatment, but this effect was attenuated in the O3MC treatment (+11%). Minor differences in the mean daily course of net assimilation were observed between the ozone and metal contamination treatments (reduction in O3/MC/O3MC by 35%/34%/46% relative to CO treatment). Assimilation showed more square daily course dynamics (Figure 5C vs. Figure 5A) in *Acer* than in *Picea*. The ozone-treated trees tended to show slightly increased levels of gas exchange compared with CO trees, because acropetal, young, and still asymptomatic leaves were measured. Net assimilation of *Acer* dropped severely in the MC (−91%) and O3MC (−82%) treatments in comparison to CO or *Picea* trees. With transpiration nearly halted, WUE in the MC/O3MC treatments increased to 209%/144% of the values measured in CO *Acer* trees. At night, the CO_2_ uptake in *Acer* trees tended to be slightly more negative in the O3 treatment than in the CO treatment, suggesting more active repair processes in the trees exposed to ozone. In parallel, a more negative transpiration value in trees exposed to metal contamination indicated higher leaf water retention.

### 2.5. Growth Reduction by O3 and Metal Stress

During the first year of the experiment, the three tested species showed contrasting growth responses to the treatments in their different organs and sometimes massive growth reduction. *Picea* was the most tolerant species, with only slightly weaker root and wood growth under metal contamination (Figure 6A; Table A3). In *Acer*, the O3 treatment caused a sizable biomass reduction, with the root/wood/foliage mass dropping to 33%/24%/24% of values measured in CO trees (Figure 6B; Table A3). The effects of MC treatment were even more severe (2%/5%/6% of values in CO trees), and they were not relieved by additional ozone stress (significant O3 × MC-interaction but no detectable antagonistic effects; Table A3). In *Tanacetum*, shoot mass tended to decrease in response to the O3 treatment, whilst root mass increased by a factor of 1.6 (Figure 6C, Table A3). Similar to the tree species, the MC treatment caused a severe biomass reduction (*Tanacetum* root mass and shoot mass = 8% and 6% of values in the CO treatment). In foliage, additional O3 stress attenuated the detrimental metal effects (significant O3 × MC interaction; Figure 6C, Table A3). The root to shoot ratio remained stable in all treated *Picea* trees (Table A3). It decreased significantly under the MC treatment in *Acer* as a result of more severe metal effects on roots than on foliage. In *Tanacetum*, the root to shoot ratio increased in response to the O3 treatment because of the aforementioned opposing biomass changes in root and foliage compartments.

During the second year of the experiment, the continuing metal exposure enhanced the biomass differences between the treated and control trees, and there were some carry-over effects of the ozone treatment from the preceding year. Whilst *Picea* biomass was not affected by the treatments during the first year, all organs of this species showed significant growth losses under metal exposure by the final harvest at the end of the second year of the experiment (root/wood/foliage mass in the MC and O3MC treatments = 46%/46%/56% and 43%/44%/55% of values in the CO plants, respectively; Figure 3A, Table A5). In *Acer*, the foliage fraction in trees exposed to ozone in the preceding year showed comparable biomass to that in CO trees. Smaller root and wood biomass (33% and 36%, respectively, of the values in the CO trees; Figure A3B, Table A5) were indicative of carry-over effects. The continued MC treatment caused stunted growth, with the dry mass in root/wood/foliage plummeting to 3%/3%/0.1% of the values measured in CO trees. In contrast to the findings from the first year, there was no change in the root to shoot ratio, irrespective of the treatment and species (Table A5).

### 2.6. Effects of Ozone Stress and Metal Contamination on the Mineral Nutrition

Mixed soil contamination with metal salts resulted in significant increases in metal concentrations in the organs of the treated plants. In *Picea*, the metal contaminants (Cd/Cu/Pb/Zn) were primarily enriched in the root organs, with concentrations 17/34/227/9 times as high, on average, as the values in CO trees (Figure 7A–D). By the end of the first year of the experiment, the concentrations of metal contaminants in roots reached levels lower than (Cu 0.4, Pb 0.3 times), similar to (Zn), or higher (Cd 1.8 times) than those measured in the surrounding contaminated soil (Table A1). In reference to the CO treatment, the wood fraction showed stronger accumulation of metal contaminants than the foliage fraction, but both were low compared to the root fraction, reaching only 3%/2%/27% of the Cd/Cu/Zn levels observed in this latter compartment (Figure 7A,B,D). In *Acer*, the responses to the treatments were mostly similar to patterns observed in *Picea* (Figure 7E–H vs. Figure 7A–D, Table A4). However, *Acer* foliage tended to accumulate more Zn, and the O3 treatment significantly (Table A4) reduced the root accumulation of Zn. The *Tanacetum* plants showed metal accumulation trends contrasting those observed in trees. Whilst the metal concentrations in the root fraction, irrespective of the treatment, reached values similar to those found in *Picea* and *Acer*, they were—apart from Cd—constitutively more elevated within shoots in plants growing in uncontaminated soil (Figure 7I–L vs. Figure 7A–H). In response to the metal treatment, *Tanacetum* showed the highest aboveground accumulation of all metal contaminants—even Pb, which generally remains blocked within tree roots. By the end of the second vs. first year of the experiment, the concentration of metal contaminants in tree organ fractions showed similar accumulation patterns. Whilst the fine root fraction showed lower Cd, Cu, and Zn concentrations than in the first year in all treatments (*Picea*; for Zn only in *Acer*), the MC treatment caused a greater Zn accumulation in foliage of both tree species.

Whilst the O3 treatment had limited effects on plant mineral nutrition, the exposure to metal contaminants significantly changed the concentration of key macronutrients within treated plant organs, with important differences between the tree and herbaceous species. In *Picea*, the C concentration in root and foliage fractions increased with the metal exposure, possibly related to drops in several macronutrients (Figure A4A–F; Table A4). Compared with trees in the CO treatment, the N/P/Ca/Mg concentration in the root fraction thus showed a 17%/32%/17%/36% reduction. Consistent with reduced gas exchange (Figure 5), the aboveground organs of *Picea* in the MC treatment had lower concentrations of phloem-unretrievable Ca compared with CO trees [54]. In *Acer*, the foliar concentration of not only Ca but also Mg was reduced under metal exposure, whereas the root concentrations of most macronutrients showed minor changes (Figure A4G–L; Table A4). Similar to metal accumulation, the macronutrient response patterns in herbaceous *Tanacetum* contrasted with those observed for trees (Figure A4M–R, Table A4). Whilst the C and Mg concentrations above and below ground were reduced by MC, N, P, and K concentrations increased, especially in foliage (Figure A4). Similar to *Acer*, there was thus an apparent correlation between higher nutrient concentration and decreased plant growth in the two species most sensitive to metal stress (Figure A4 vs. Figure 6). At the end of the second vs. first year of the experiment, the macronutrient concentrations in the organs of the two tree species showed similar responses to the treatments, with foliar Mg reduction in response to MC treatment becoming significant for *Picea*, corresponding to the aforementioned greater needle discolouration.

Despite having contrasted ecological strategies (plant economic spectrum, [55,56], with *Picea* being more conservative (slow-return) and *Acer* and *Tanacetum* being more acquisitive, untreated (CO) plants of the three species showed rather similar fitness and performance levels. Given the optimal soil and climate conditions, the current-year foliage of *Picea* had a higher LDMC than that of *Acer* (Figure 2B vs. Figure 2G). Regarding nutrients, the root and foliage fractions of *Picea* had lower K, Ca, and Mg, higher N, and similar P concentrations compared with *Acer* and *Tanacetum* (Figure A4). Day-time average gas exchange levels were similar in Picea and Acer (Figure 5).

## 3. Discussion

### 3.1. Stress Responses to Ozone and Metal Contaminants

Visible injury caused by ozone and by metal stress primarily differed as a consequence of the stomatal (ozone) vs. vein (metal) penetration route of the stress agent within foliar tissues, irrespective of the species. The most frequent symptom was a homogeneous degradation of foliar green leaf colour, which was observed in all species and treatments. This injury symptom is a typical ACS marker [31], missing specificity and with low injury explanation power [57]. It provided an early stress indication in the first year of the experiment [28] and denoted a carry-over effect of first-year ozone stress during the second year of the experiment. In the tree and forb species, visible ozone injury appeared with AOT40 = 9–13 ppm·h, suggesting low ozone sensitivity [29,58,59]. Visible injury in *Acer* was similar to that previously reported in the case of ozone [29] and metal [36] stress, whereas stress responses of *Tanacetum* are reported here for the first time. In *Biscutella laevigata*, homogeneous foliar discoloration in response to metal stress, as observed here for the two angiosperm species, was typically observed in the non-metallicolous genotypes, and contrasted with more specific—i.e., mottling—symptoms in metal-adapted plants [38]. Chlorophyll Mg substitution by Zn, in addition to ACS processes—as indicated by lowered N concentrations in *Acer*—could contribute to foliar discolouration [60]. This was particularly the case regarding *Tanacetum*, where Mg was the only soil macronutrient showing decreased concentrations by MC. Whilst induced by both stress factors, premature shedding in response to the MC treatment of just the preceding-year needle generation in *Picea* could have protective effects, reducing intoxication of growing tissues, e.g., by Zn recycling from older and more contaminated foliage [33].

Structural injury by ozone stress was primarily observed within mesophyll cells, whereas the vein tissues remained unharmed. The most characteristic changes were found in chloroplasts, as a consequence of their sensitivity to oxidative stress [61] and overload of the antioxidant system [62]. The chlorophyll degradation, poor thylakoid resolution, and enhanced accumulation of plastoglobuli observed in *Acer* and *Picea* were thus indicative of injury and repair mechanisms [28,63]. Synergistic light stress effects were denoted by gradients of chlorophyll degradation increasing in severity adaxially within the mesophyll and upper palisade cells of *Tanacetum* and *Acer*. Similarly, the light-exposed side of *Picea* upper twigs showed photobleaching. Chloroplastic changes were part of the typical ACS cell responses to treatments—also including enhancement of vacuome size, condensation of cytoplasm and nucleus, and the accumulation of condensed tannins [64]. Using the vanillin-acid reagent, the red-stained oligomeric tannin fractions (OPC)—with higher antioxidant properties [65,66]—showed up as solutes in the vacuoles. The solid tannin bodies formed by tannin polymers (PPC; [28]), showed an atypical brownish reaction; heat and stronger acids are needed for specific PPC histochemical staining [27,67].

A main difference regarding structural injury by metal vs. ozone stress concerned the tissues affected in priority within foliar organs. In *Acer* foliage, gradients of defence and stress reactions under metal stress spread from leaf veins towards surrounding lower leaf blade tissues. They indicated the import pathways for mobile metal contaminants (Cd, Zn), which extend along the main apoplastic and symplastic water and nutrient routes [34,68]. Conversely, besides discrete accumulations within upper mesophyll cell layers, mobile contaminants (Zn and Cd) are generally detected within the lower epidermis and mesophyll [35,36,51]. The co-accumulation of tannin oligomers therein can contribute to the scavenging for oxygen radicals and other ROS triggered by metal stress [51,66,69]. These phenolics are not involved in the complexation and detoxification of mobile heavy metals [35,36,51]. Consistent with an accumulation in response to ozone or many other environmental constraints [27], tannin production thus formed a rather baseline stress reaction, in association with unspecific ACS processes. In relation to species-specific contaminant uptake and mobility, Zn contaminants were accountable for the metal stress in the aboveground organs of *Picea* and *Acer*, whereas all four metals could affect the foliage of *Tanacetum*.

Condensed tannins represented the main phenolic fraction and certainly the most frequent defence compounds elicited in material from tree plants and in response to ozone and metal stress. In root (*Picea*) and foliage (*Acer*) organs, single OPC and PPC tannin fractions exceeded 3% of the organ dry mass in some cases. Their biomass contribution was thus slightly lower (about −1%) than in beech and spruce foliage under elevated CO_2_ and nitrogen deposition, treated in ambient conditions [70], or in mountain birch (i.e., OPC + PPC = 14% dry mass [71]). With such sizable levels, tannins could contribute to LDMC increases in foliar organs (*Acer*), in addition to increasing the cell wall thickness. The accumulation of condensed tannins provided an organ-based indication of relative stress intensity. In foliage, while larger OPC and PPC fractions in ozone-treated *Picea* trees indicated more severe stress, soilborne pollution elicited larger defence reactions in *Acer*. Metal contaminants were also the primary cause of tannin accumulation in *Picea* roots, further confirming the driving role of metal allocation regarding defence reactions. Hence, the MC treatment caused more detrimental effects at the whole plant level than the O3 treatment, irrespective of the species. The solubility of the applied metal salts seemed to be primarily accountable for this observation, considering findings obtained in hydroponic conditions, where stunted growth has also been reported [72]. In ambient conditions, only a fraction of metal contaminants is thus available to plants [73]. The lower abundance of cell-wall-bound PPC within stressed roots of *Picea* might indicate increased fine root turnover in these highly plastic organs [74]. Such a belowground process was further suggested by the decreased phenolic oxidation intensity, which showed a striking apparent negative correlation with PC accumulation.

### 3.2. Changes in Functional Traits within Foliage in the Context of Species-Specific Ecological Strategies

The responses to the treatments of functional traits in foliage showed marked interspecific differences. In *Picea*, higher LMA and WUE, reduced gas exchange, and lowered P (but not N) needle concentrations in response to the O3 treatment were indicative of acclimation and an enhanced conservative strategy as a consequence of the more constraining environmental conditions [55,75]. Higher LMA could result from thickened cell walls and increased tannin concentrations. Together with enhanced WUE, it might then offer conifers better protection against oxidative stress. Shedding of preceding-year needles represents a typical reaction of non-acclimated foliage to experimental exposure [30,31]. In our experiment, marked drops of LMA and gas exchange in *Acer*, especially in response to the MC treatment, represented clear stress reactions. These effects conferred no apparent stress tolerance and appeared unrelated to the species’ growth strategy. Enhanced LDMC in response to metal stress could relate to blocked gas exchange and halted transpiration, resulting from injury along the water transport pathways. Metal stress has been observed to decrease leaf water content [76] and reduce gas exchange [37,77]. Hence, the responses of functional traits to MC treatment in our study provided evidence for the exceedance of the stress tolerance threshold in *Acer* trees. Slightly higher gas exchange levels in young and still asymptomatic leaves under ozone stress contrasted with findings based on mature foliage [30,78,79]. This difference may relate to the unresponsiveness of still-developing foliage [31] or to compensation mechanisms [58].

### 3.3. Contrasting Effects of Ozone and Metal Stress on Biomass and Mineral Nutrition

The biomass responses to the treatments depended on the species-specific ecological strategy. As a species with a “slow-return” strategy [55,56], *Picea* showed high stress tolerance, with sizable biomass reduction only after 2 years of MC treatment. However, it still showed isometric scaling, with conserved organ size relationships indicative of further growth potential under severe environmental stress, if the experiment had been prolonged. Under exposure to environmental stress, species adapted to limiting environmental conditions may retain normal growth in the short term, while showing visible injury [80]—with underlying enhancement of conservative functional traits, as in our case. Significant biomass loss in the longer term may result from reduced photosynthetic carbon gain [81]. With a more acquisitive strategy, *Acer* was the most sensitive species out of the three tested, with considerable biomass loss in response to both stress factors. Root and wood biomass reductions in the second year, in trees previously exposed to ozone stress, indicated a so-called memory effect [82]. With a lower ozone dose than in the present study, other *Acer* species have also shown decreased growth [78]. However, the allometric relationships in trees from the O3 treatment were conserved and the roots and other organs were affected in a similar way by the one-year exposure, in contrast to findings for other forest seedlings from the northern hemisphere [83]. This result suggests possible higher ozone stress tolerance in the selected *Acer* species, although genotypes may also show varying tolerance to ozone stress, as shown for birch [84]. By contrast, metal toxicity in the MC treatment caused massive root growth limitation, similar to the response of *Tanacetum*, and then a reduced root to shoot ratio in *Acer*, as observed for crops [46]. Similar to this previous evidence, the allometric effects in *Acer* were transient and the blocked root system development caused isometric stunted growth in the second year, as observed with exposure to elevated Cd concentrations and high metal availability [72].

The MC treatment caused sizable and comparable increases in metal concentrations in the roots of all three tested species. By contrast, large interspecific differences were observed in the amounts translocated into the shoot and foliage tissues after crossing the root endodermis filter; still, most contaminants remained blocked in the outer absorbing root tissues, particularly within metal-treated trees. In the two tree species, only Zn was translocated into foliage at levels exceeding the lower toxicity level for various plants [85]. However, in the *Tanacetum* ruderal, all four contaminant concentrations increased to substantial amounts above ground, in contrast to usual metal translocation patterns [32]. Cd and Zn even reached higher concentrations than in roots and therefore exceeded the toxicity threshold. This finding outlines the possible dissemination threat posed by spontaneous ruderal vegetation at e.g., brownfield sites, when primarily a phytostabilisation effect is targeted [47]. Zn contaminants retranslocated to nutrient stores over the winter [33] could have contributed to the greater Zn accumulation in the second year of the experiment compared with in the first year.

Despite some significant nutrient reductions, the mineral nutrition of the two tested tree species did not show deficiencies under metal exposure. Only Mg in *Acer* foliage showed concentrations below the sufficiency range, which could result from uptake competition with excess Zn [86]. Nutrient deficiencies in the vegetation that reclaims brownfield sites [87] may thus primarily result from otherwise dystrophic site properties, rather than from direct effects of metal contamination. Further confirmation was provided by the increased nutrient concentrations in *Tanacetum* plants under metal exposure, which probably resulted from the severely decreased biomass accumulation.

### 3.4. Multiple Airborne and Soilborne Stress Factors

Irrespective of the species, plant organ, or assessed variable, we observed few significant interactions between the O3 and MC treatments. Moreover, they generally resulted from detrimental ozone stress effects in plants exposed to this single experimental factor, reducing the response gap with plants in the combination treatment—where reactions were primarily driven by metal stress. Notably, such attenuation effects were observed in the case of biomass reductions in *Acer*. In *Picea*, they were found for the shedding of preceding-year needle biomass and root tannin concentrations. Antagonistic ozone and metal stress effects were observed for the current-year leaf colour index (*Picea*) and LMA (*Acer*) only; no additive stress effect was ever observed. Findings from the limited number of comparable studies also indicate varying stress factor interactions, also given peculiarities in the experimental designs. With exposure to Cd contamination, followed by a single pulse with high ozone concentration, both stress factors depressed the foliar CO_2_ assimilation of *Helianthus annuus*, but the effects were not additive and changes in the rubisco activity indicated that Cd was the main stress factor [88]. In other studies with this same treatment combination, only the ozone exposure reduced the photosynthetic capacity in poplar and tomato [89,90]. In addition to species and experimental design factors, additive, antagonistic, or a lack of interactions to combined Cd and ozone stress were also observed to depend on the measured parameters [91,92], as in our case. The overwhelming toxicity of MC treatment probably contributed to the rarity of interactions between the stress factors in our study. Sequential application and moderate stress severity can trigger “hardening” and cause so-called cross-adaption in the treated material, as observed with ozone or drought pre-treatment of barley seedlings, which contributed to enhancing catalase activity and reducing oxidative stress and lipid peroxidation under Cu (but not Cd) stress [93]. Similarly, consistent enhancement of plant defences was observed in more tolerant *Picea* plants in our study, suggesting, together with a larger C investment in roots and foliage, further strengthening of its conservative, “slow-return” strategy. Whilst the responsiveness varied according to organ and treatment, the response pattern under both stress factors remained the same, and these stress-unspecific responses could have contributed to the observed tolerance to both stress factors. Given their high tolerance and integrative responsiveness to environmental stress, species with conservative ecological strategies thus appear to be well suited to managing metal-polluted sites when phytostabilising effects are targeted—also given the rising environmental constraints in the context of ongoing climate change.

## 4. Materials and Methods

### 4.1. Experimental Design and Treatments

The facility at the Swiss Federal Institute for Forest, Snow and Landscape Research WSL (N 47°21″43″/E 8°27′24″, 550 m a.s.l.) used during the first year of the experiment consisted of two large (10 m^2^) walk-in growth chambers, enabling full control and close replication of environmental and ozone conditions measured from 14 April to 30 September (1998) at an ozone pollution hotspot in Switzerland (Morbio Superiore, N 45°51′41″/E 9°01′47″, 599 m a.s.l.; Figure A1). The temperature and humidity were adjusted every 15 min to correspond to the values at Morbio Superiore during the reference period. The daylight course was simulated by adjusting the light intensity of 25 lamps per chamber every hour (Power Star HQI-E 1000 W, Osram, Munich, Germany). The control treatment (CO) chamber received a background ozone concentration of 20 ppb added to filtered air (charcoal and PuraFilter^®^, Purafil, Doraville, GA, USA), which was supplied at a rate of 100 m^3^ h^−1^ with a wind speed of 0.5 m s^−1^. The ozone treatment (O3) chamber was supplied additionally with the ozone concentrations found at Morbio Superiore (during the reference period, adjusted every 15 min using a computer-assisted mass flow controller. Ozone was generated from pure oxygen by electrical discharge, using an Ozone Generator 500 M (Thermo Fisher Scientific, Bremen, Germany), with concentrations monitored using an Ozone Analyzer Model 8810 (Teledyne Englewood, CO, Labs, Inc., Englewood, CO, USA). By the end of exposure, the concentration-based ozone dose (AOT40) reached 35 ppm·h.

The plant material was overwintered in field conditions at WSL. During the second year of the experiment, it was kept outside under ambient climatic conditions but with the metal treatment still in effect (see below). The ambient environmental conditions and ozone pollution were monitored between April and October using an automated weather station, with air temperature, humidity, precipitation, and ozone concentrations measured every 15 min at 2 m above ground level and averaged per hour.

The soil substrate, with balanced nutrient supply (Table A1), consisted of slightly acidic silty loam from an arable field (Birr, Aargau, Switzerland). It was spiked with elevated levels of several heavy metals (Cd/Cu/Pb/Zn; Table A1) to establish the metal contamination (MC) and combination (O3MC) treatments. The soil pollution was achieved by adding the metals to the soil substrate in the form of sulphate/pentahydrate salts dissolved in water, such that concentrations exceeded the Swiss soil remediation values [94]. The same amounts of sulphates were added to the CO treatment. In reference to European critical limits, the values exceeded the EU ecological risk threshold/EU standards by a factor of about 1.5/10 (Cd), 5/7 (Cu), 2.5/7 (Pb), and 5/7 (Zn) [19,48]. By the final harvest (end of the second year of the experiment), only a minor decrease in the heavy metal contamination was detected, restricted to the pots with *Acer pseudoplatanus*. Watering during the experiment was performed from below, using subplates to prevent leaching. For each species, plants (n = 40 and 20 for trees and forbs) were randomly assigned to either contaminated or uncontaminated pots. The two plot cohorts per species were then randomly halved between the CO and O3 treatment chambers in the experimental facility and further randomised by pot position within each chamber (each with 50 pots) to achieve the targeted four treatments (chamber 1: CO, MC; chamber 2: O3, O3MC).

The selected plant material belonged to three species typical of afforestations and spontaneous vegetation at brownfield sites in Central Europe and representative of dominant plant life forms: two megaphanerophytes and one hemicryptophyte, namely a coniferous tree (Norway spruce; *Picea abies* (L.) Karst.), a deciduous tree (sycamore; *Acer pseudoplatanus* L.), and an herbaceous ruderal (tansy; *Tanacetum vulgare* L.). Plant material, consisting of five-year-old *Picea* seedlings (average biomass and size: 49.0 ± 1.4 g and 28.7 ± 3.7 cm), one-year-old *Acer* rooted cuttings (3.7 ± 0.1 g and 28.2 ± 3.9 cm), and 10-cm-long *Tanacetum* rhizome segments produced using local provenances, was supplied by the WSL horticultural facility. All material was planted into 10 L pots individually whilst still dormant, then moved to the experimental facility, where flushing occurred two months later.

### 4.2. Visible, Morpho-Anatomical, Biochemical, and Physiological Assessments

The occurrence of symptoms in foliage was assessed visually once a week during the entire vegetation season in the first year of the experiment. By 25 September, some 100 days after flushing, the foliar injuries caused by ozone and heavy metal stress separately and in combination were assessed in five plants per treatment, selecting lateral twigs in one branch from the second highest whirl (*Picea*) or five representative leaves per plant in terms of leaf size, age, and visible symptom spectrum (*Acer and Tanacetum*). The average pigmentation of leaves or needles (current and preceding-year needle generation) was estimated using colour charts [95], converting the readings into a semi-quantitative 0 to 10 (yellow-brown to dark green) rank variable. Tree leaf morphology responses were assessed as leaf dry matter content (LDMC) and leaf mass per area (LMA) from the material used for the foliage colour assessments [70].

Injuries at the tissue and cell level underlying the observed stress symptoms in the two tree species were investigated by means of structural and histochemical observations using light (LM), fluorescence (FM), and transmitted electron (TEM) microscopy. The degradation of chlorophyll and accumulation of cell wall phenolics in current-year foliage was observed under UV-light excitation (FM; [29]) using 60-µm-thick unfixed hand-microtomed sections. Proanthocyanidins (PCs) were observed histochemically, using similar preparations and the vanillin acid test derived from [96] and [29]. All observations were performed using the 5× to 100× objectives of a Leitz DM/RB microscope (Leica Microsystems, Heerbrugg, Switzerland), and micrographs were taken with the analogous micrograph system in use at that time (Wild MPS 48/52, Kodak Ektachrome 400 ASA films). For further cytological observations using TEM, 1-mm^2^ pieces of fixed (2.5% buffered glutaraldehyde) leaf and fine root samples (<2 mm) collected during the first-year harvest (see below) were rinsed twice in 0.067 M phosphate buffer, post-fixed in 2% buffered osmium tetroxide (4 °C), dehydrated in ethanol, and embedded in a PolyBed-Araldite epoxy resin mixture. Ultrathin 90-nm sections were trimmed using a LKB Ultratome III (LKB Stockholm, Sweden) and post-stained in uranyl acetate and lead citrate prior to observation in a JEOL JEM-1010 (JEOL Inc. Peabody, MA, USA) microscope [97].

The induction of plant defences was analysed by measuring several PC fractions and polyphenolic properties in current-year foliage (*Picea* and *Acer*) and fine roots (*Picea*), using plant material sampled at the end of the first treatment year (see below). Foliar and fine root aliquots were shock frozen, freeze-dried, and analysed according to [70]. Briefly, 1 g of freeze-dried material was homogenised prior to the extraction and purification of phenolics according to [98]. The intensity of phenolic oxidation was determined according to [99]. PC fractions were quantified using the acid-vanillin (mainly the oligomers, OPC, [98,100]) and proanthocyanidin (mainly the polymers, PPC; [100,101]) assay. This latter assay was also used to quantify PC in the insoluble and primarily cell wall fraction (PPCcw). The absorbances were read on a UV-160 spectrophotometer (Shimadzu, Kyoto, Japan), and the results were expressed as (+) catechin (acid-vanillin assay) and perlargonidin (PC assay) equivalents.

The physiological responses to the treatments in tree foliage [102] were characterised by measuring daily courses (four measurements per hour) of leaf gas exchange for 32 days, selecting three trees per species and treatment (*Picea:* from 23 August to 23 September, AOT40 increase from 17 to 22 ppm·h; *Acer*: 31 July to 31 August, AOT40 increase from 11 to 14 ppm·h). All measurements were performed in situ using lateral current-year shoots (*Picea)* or top leaves (*Acer*), a Walz2 CQP130 gas-exchange system (Walz, Effeltrich, Germany), and a Li-Cor2 6262 infrared gas analyser (Li-Cor, Lincoln, NE, USA), measuring in parallel but alternating the assessed treatments daily (to homogenise gas-exchange device error).

### 4.3. Biomass and Chemical Assessments

In the first harvest, at the end of the first year of the experiment, half of the trees (n = 20 per species and treatment), all tansy plants, and all shed foliage were collected. The remaining trees were then transferred to the WSL horticultural facility, where they were left for one year in ambient field conditions, until a second harvest at the end of the second year of the experiment. At each harvest, the collected material was divided into soil (tree pots only), root (cleaned coarse and fine roots < 2 mm diameter), wood (tree stems and twigs), and foliage (including the herbaceous shoots of tansy plants) fractions, and the CaCl_2_ pH of the soils was determined. All plant material was oven-dried at 65 °C until a constant weight was reached, and dry mass was assessed. The material within each fraction was then homogenised, subsampled for elemental analysis, and ground to a fine powder (Retsch MM2000 zirkonoxid-bowl ultra-centrifuge mill, Retsch GmbH, Hann, Germany). It was then digested in a high-pressure microwave system (240 °C, 12 MPa; ultraClave, Milestone, Sorisole, Italy) and analysed in duplicate (spread < 10%) using a gas chromatograph (NC-2500, Carlo Erba-Instruments, Wigan, UK) for C and N and by ICP-OES (Optima 7300 DV, PerkinElmer Inc., Waltham, MA, USA) for Cd, Cu, Pb, Zn, and the macronutrients Ca, K, Mg, and P according to ISO 17025.

### 4.4. Statistical Analysis

In a first series of analyses, the effects of treatments, species, organs (split-plot design, with the subplot organ factor nested in the species factor), and two-way interactions on the foliar morphology, foliar and root tannins, organ biomass, and chemical element concentrations in the material harvested after the first year of the experiment (26 response variables) were analysed using general linear models (GLM) and their significance tested by means of ANOVA (type III sum-of-squares, using log-transformed data). Given the nearly always significant species and organ effects, the statistical analyses were repeated taking each species and plant organ separately (Table A2, Table A3, Table A4 and Table A5). Biomass and chemical data from the second assessment year were analysed separately, as measurements were performed using plants left after the first harvest (= no true measurement repetitions). In the case of occasional chemical element concentrations below the detection limit (dl), half the dl value was assigned [103]; groups of data with > 60% of their values below the dl were excluded from statistical analysis. In the case of leaf colour data (rank variable), the statistical results were verified using nonparametric tests. Differences between group averages were tested by means of pairwise Tukey’s studentised range (HSD) tests. All statistical analyses were performed using the SAS software package (release 9.4, SAS Institute Inc., Cary, NC, USA). All figures were plotted using non-transformed data.

## 5. Conclusions

Findings from this study outline the contrasts and similarities regarding the effects of ozone and metal stress factors and the importance of species-specific ecological strategies in plants reclaiming brownfield sites, with respect to the observed plant responses and stress tolerances. In all cases, the foliar or root organ directly exposed to the airborne or soilborne stress agent was also the organ that was primarily impacted (confirmation of hypothesis 1a). However, both stress agents also affected other organs that were not directly exposed, most likely due to the affected assimilation and mineral nutrition processes. In most cases (26 parameters observed), there were no interactions between the two stress factors, or such an effect resulted from a smaller injury difference between plants under ozone vs. combined stress exposure compared with those under the control vs. combination treatment (rejection of hypothesis 1b). Strikingly, no additive stress effect was ever observed, and the few antagonistic interactions or hardening effects suggest the prevalence of integrative stress responses, enhancing “slow-return” traits, especially in species with conservative strategies. Ozone stress memory effects in the second year were suggestive of additional environmental influences by transient stress factors, which—in addition to elevated ozone concentrations in a sunny year—may include drought stress. Given the observed detrimental effects on perennial root and wood organs of *Acer*, the stress-enhancing vs. hardening aspects of such memory effects require further research. Both stress agents elicited similar types of responses, especially within the same plant life form. However, there were differences between species, notably regarding the contaminant concentrations in shoots, which were much more elevated in the ruderal species than in the corresponding organs of treated trees (partial confirmation of hypothesis 2a). The ruderal vegetation at contaminated sites may thus pose a remobilisation and dissemination threat regarding soil contaminants, especially when a phytostabilisation effect is targeted. Stress severity was highly species-specific, with the strongest effects overall being observed in *Acer*, while *Picea* showed higher stress tolerance (confirmation of hypothesis 2b). Lower rates of gas exchange under ozone stress or reduced contaminant translocation to foliage in the MC treatment suggest that this better tolerance was primarily achieved by excluding the stress agent. Enhanced defence reactions increasing ROS scavenging activity, together with cell wall thickening, could further contribute to reduced injury. Comparing findings between compartments, large reductions in biomass, and increases in tannin concentrations in roots or a changed root to shoot ratio suggest a tendency of higher stress severity belowground (partial confirmation of hypothesis 2c). This may be related to the high availability and prevailing stress effects of applied metal salts. Integrative stress responses and the absence of additive stress effects suggest overall good mechanistic tolerance to multiple stressors in the treated species. With the current increases in environmental stress, our findings outline the relevance of “slow-return” species strategies, with low productivity but enhanced stress tolerance. Their low nutrient requirements also appear of interest in several instances for managing brownfield sites if confinement of environmental pollution through phytostabilisation is targeted.

## Figures and Tables

**Figure 1 plants-12-03011-f001:**
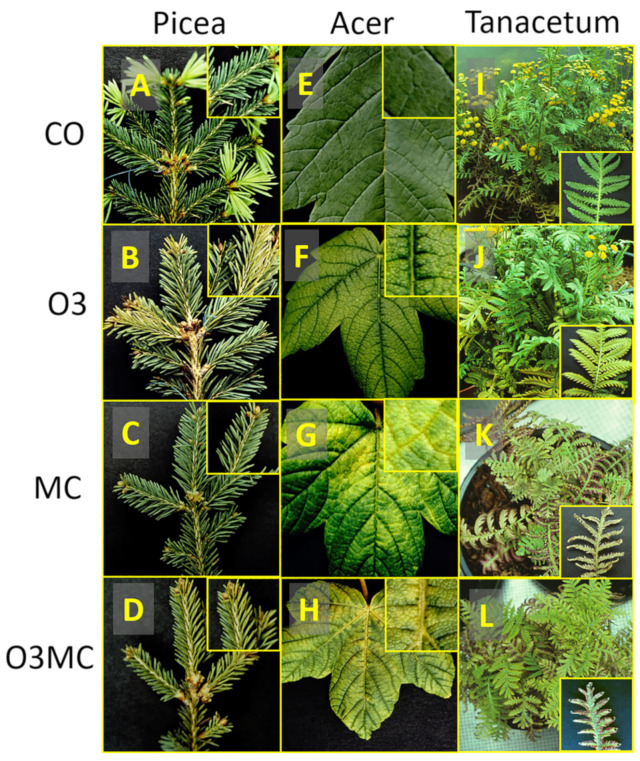
Exemplary visible symptoms in response to elevated ozone (O3, AOT40 = 35 ppm·h), metal contamination (MC), and the combination of the two treatments (O3MC) vs. control treatment (CO), as observed in foliage of *Picea abies* (**A**–**D**), *Acer pseudoplatanus* (**E**–**H**), and *Tanacetum vulgare* (**I**–**L**) by the end of the first year of the experiment (100 days after flushing). Second flush needles in *Picea* were observed in the control treatment only. Flowering of *Tanacetum* was reduced by elevated ozone and prevented by metal contamination.

**Figure 2 plants-12-03011-f002:**
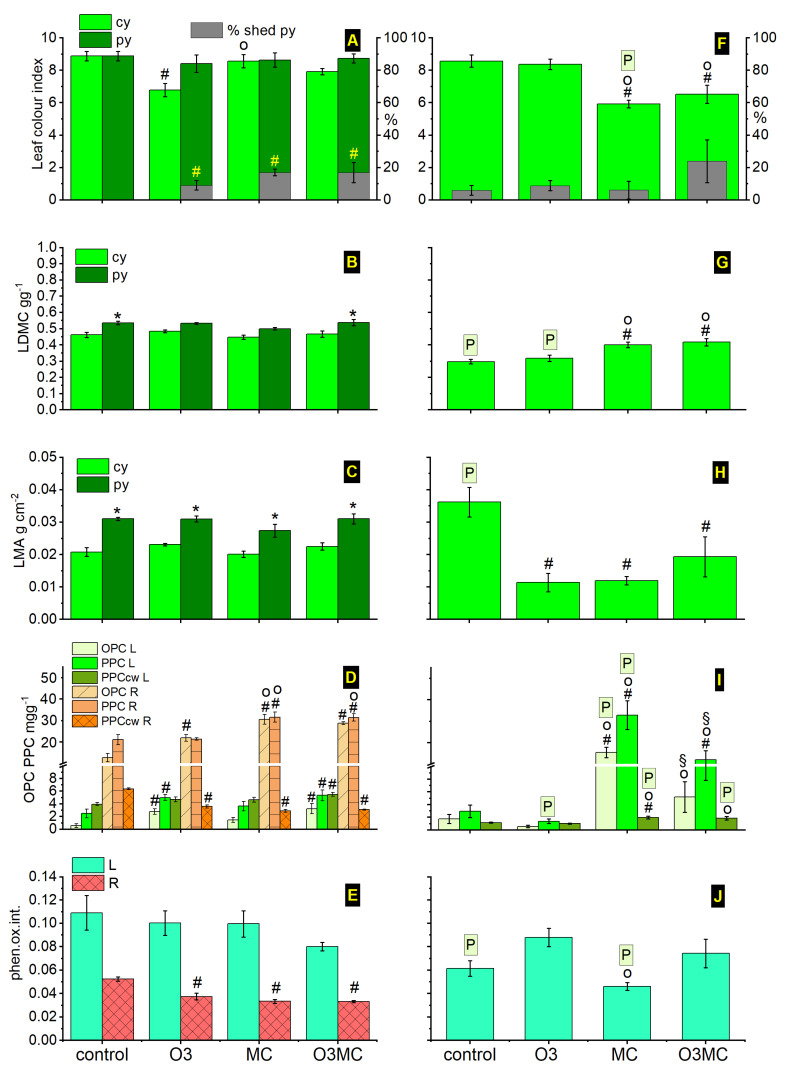
Changes in foliar (**A**–**J**) and fine root (**D**,**E**) traits of *Picea abies* (**A**–**E**) and *Acer pseudoplatanus* (**F**–**J**) in response to elevated ozone (O3: AOT40 = 35 ppm·h), metal contamination (MC), and the combination of the two treatments (O3MC) vs. control treatment (CO), as observed by the end of the first year of experiment (100 days after flushing); mean values ± SE, n = 5 plants. The condensed tannin fractions are expressed as (+) catechin (OPC) and pelargonidin (PPC, PPCcw) equivalents. Abbreviations: current-year (cy) and preceding-year (py) needle generations; leaf dry matter content (LDMC) and leaf mass per area (LMA); oligoproanthocyanidins (OPC), cellular polymerised proanthocyanidins (PPC), polymerised proanthocyanidins within cell walls (PPC_cw_), phenolic oxidation intensity (phen. ox. int.), L = leaves, R = fine roots. Significance (*p* < 0.05 pairwise Tukey’s studentised range test comparisons): significantly different from control (#), O3 (o) and MC (§) treatment, from *Picea* same treatment (framed P), and from current-year needle generation (*). See Table A2 for the significance of the experimental factors.

**Figure 3 plants-12-03011-f003:**
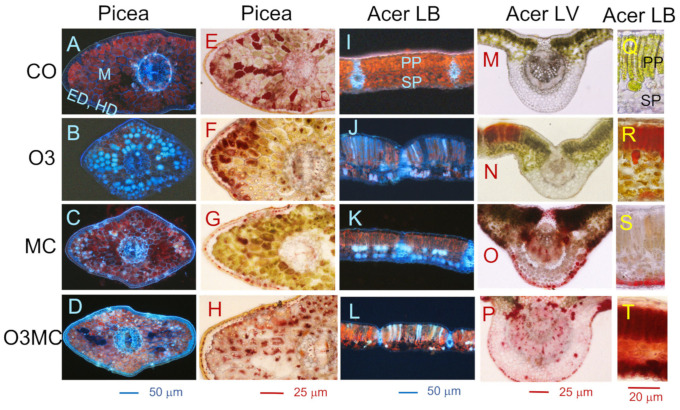
Histochemical changes underlying visible injuries in foliage tissues of *Picea abies* (**A**–**H**) and *Acer pseudoplatanus* (**I**–**T**) in response to elevated ozone (O3), metal contamination (MC), and the combination of the two treatments (O3MC) vs. control treatment (CO). (**B**–**D**,**J**–**L** vs. **A**,**I**): degradation of chlorophylls (reddish) and accumulation of polyphenolics (bluish) in cell walls of outer epidermal (ED) and hypodermal (HD) needle tissues and vacuoles within mesophyll (M), palisade (PP), and spongy (SP) parenchyma. Metal contamination caused more phenolic accumulation in cell walls, especially in outer needle and lower leaf blade tissues (**C**,**D**,**K** vs. **A**,**I**). (**F**–**H**,**N**–**P**,**R**–**T**) vs. (**E**,**M**,**Q**): accumulation of proanthocyanidin oligomers (OPC, reddish) in vacuoles of needle mesophyll and leaf blade (LB) and leaf vein (LV) tissues. OPCs accumulated in palisade parenchyma with ozone stress but in lower leaf blade and vein tissues with metal contamination (accumulation in both locations in the combination treatment). Technical specifications: unfixed fresh sections visualised under UV-light excitation (**A**–**D**,**I**–**L**) or with bright field microscopy (**E**–**H**,**M**–**T**) after OPC revelation using the vanillin acid test.

**Figure 4 plants-12-03011-f004:**
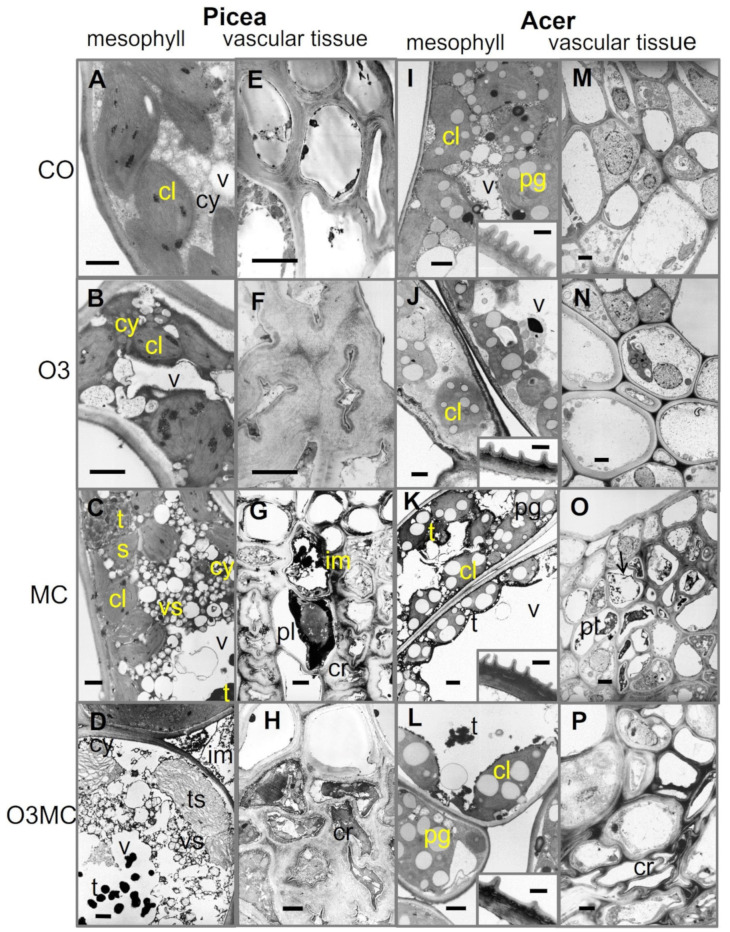
Degenerative changes in the cell structure of mesophyll and vascular tissues underlying visible injuries in foliage of *Picea abies* and *Acer pseudoplatanus* in response to elevated ozone (O3 treatment), metal contamination (MC treatment), and the combination of the two treatments (O3MC) vs. control treatment (CO; **A**,**E**,**I**,**M**). In response to the O3 treatment, the cell cytoplasm (cy) and chloroplasts (cl) showed increased condensation (**B**,**J** vs. control **A**,**I**), while the vascular tissues remained asymptomatic (**F**,**N**). The MC and O3MC treatments showed stronger injuries (**C**,**K**,**D**,**L**), with additional leakage of cell debris into the intercellular space (im), cytoplasm vesiculation (vs), accumulation of tannins (t, black) in the vacuole (v), apparent increase in size and frequency of plastoglobules (pg; *Acer*), or thylakoid swelling (ts; *Picea*). Within the phloem, plasmolysis (pl) and cytorrhysis (cr) suggested necrosis (**G**,**O**,**H**,**P**). In *Acer* leaf blades, the outer cell wall and cuticular ridges of epidermal cells (inset **J**,**K**,**L**) showed inlays of what is probably phenolic material, according to biochemical and histochemical evidence (Figure 2I and Figure 3K). Technical specifications: postfixation using OsO_4_, contrasting using uranyl acetate and lead citrate, observation in transmission electron microscopy (TEM). Bars = 2 µm.

**Figure 5 plants-12-03011-f005:**
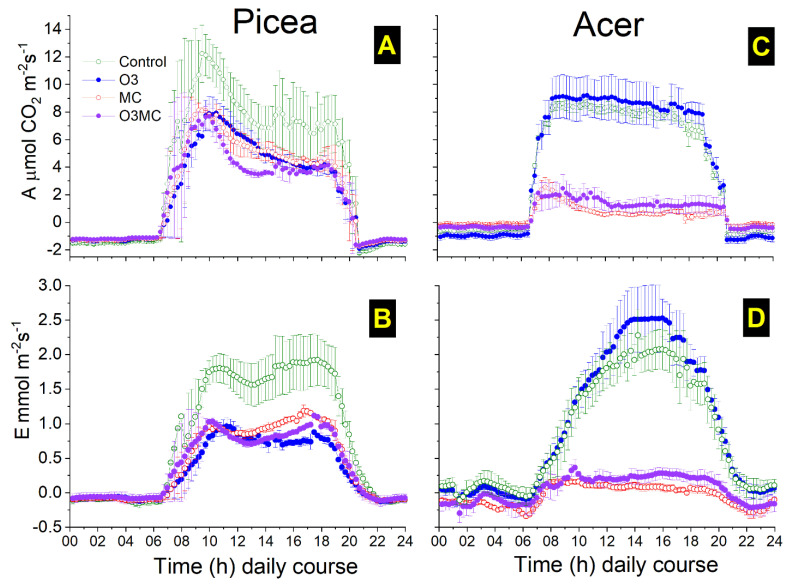
Average daily course of leaf gas exchange (**A**,**C**: A_area_ net CO_2_ assimilation; **B**,**D**: transpiration) in response to elevated ozone (O3), metal contamination (MC), and the combination of the two treatments (O3MC) vs. control (CO) treatment (mean values ± SE, n = 3 trees). The measurements were performed from 23 August to 23 September (*Picea abies*) and from 31 July to 31 August (*Acer pseudoplatanus*), with the ozone dose (AOT40) increasing from 17 to 22 (*Picea*) and from 11 to 14 (*Acer*) ppm·h in the ozone treatment chamber during that period (Figure A1E). All daily course measurements (4 per hour) were realised in situ, using lateral current year shoots (*Picea*) or top leaves (*Acer*).

**Figure 6 plants-12-03011-f006:**
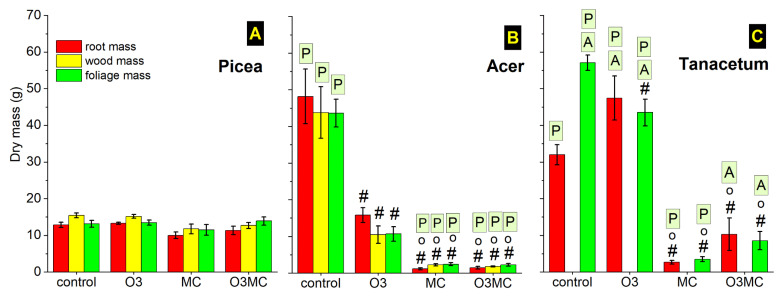
Changes in the total root, wood, and attached foliage dry mass of *Picea abies* (**A**), *Acer pseudoplatanus* (**B***)*, and *Tanacetum vulgare* (**C**) plants by the end of the first year of the experiment in response to elevated ozone (O3), metal contamination (MC), and the combination of the two treatments (O3MC) vs. control (CO) treatment (mean values ± SE, n = 5 plants). Significance (*p* < 0.05 pairwise Tukey’s studentised range test comparisons): significantly different from CO (#) and O3 (o) treatment (n.s. differences between MC and O3MC); significantly different from *Picea/Acer* (framed P/A), same treatment. See Table A3 for the significance of the experimental factors.

**Figure 7 plants-12-03011-f007:**
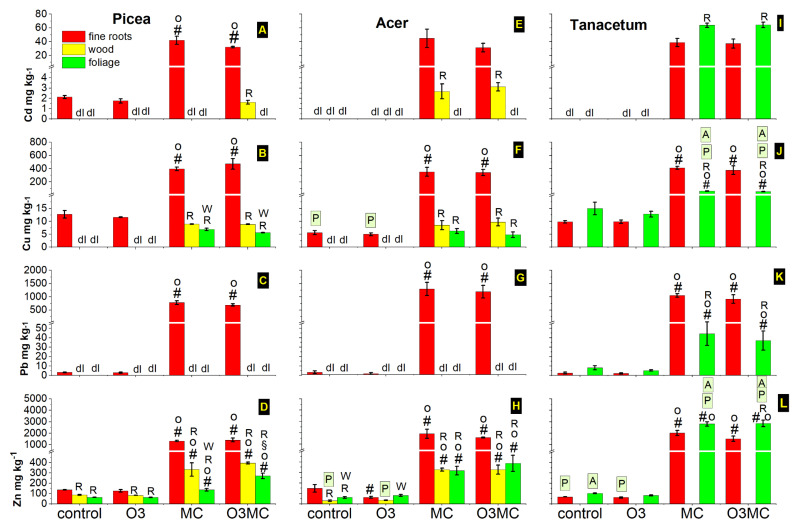
Changes in the concentration of metal contaminants in the fine root, current-year wood, and foliage fractions of *Picea abies* (**A**–**D**), *Acer pseudoplatanus* (**E**–**H**), and *Tanacetum vulgare* (**I**–**L**) plants by the end of the first year of the experiment in response to elevated ozone (O3), metal contamination (MC), and the combination of the two treatments (O3MC) vs. control (CO) treatment (mean values ± SE, n = 5 plants). Detection limit (dl) for Cd/Cu/Pb = 1.5/4.5/1.5 mg kg^−1^. Significance (*p* < 0.05 pairwise Tukey’s studentised range test comparisons): significantly different from CO (#), O3 (o), and MC (§) treatment from the root (R) and wood (W) fraction within the same species and treatment, and from *Picea/Acer* within the same organ and treatment (framed P/A). See Table A4 for the significance of the experimental factors.

## Data Availability

Data are available upon request by contact with the corresponding author.

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
