# Peer review of "Responses to Airborne Ozone and Soilborne Metal Pollution in Afforestation Plants with Different Life Forms"

_plants, 2023, doi:10.3390/plants12163011_

Round 1

Author Response

REVIEWER #1

“Possible negative effects of ozone and metal contamination on plants determine the relevance of the reviewed work. In it, on a large amount of factual material, it was shown that the foliar or root organ directly exposed to the airborne or soil borne stress agent was also the organ that was primarily impacted. However, both stress agents also affected other organs that were not directly exposed, most likely due the affected assimilation and mineral nutrition processes.

Particularly attractive, in our opinion, is that the author of the peer-reviewed paper showed no differences between species, notably regarding the contaminant concentrations in shoots, which were much more elevated in the ruderal species than in the corresponding organs of treated trees. This study can be considered to be the great contribution to the treasury of publications of plants (trees) cultivating in urbanized areas.

In terms of the volume of the presented results, their theoretical and practical significance, the study can be recommended to be published in concerned journal.”

OUR ANSWER: Thank you very much for the overall positive appreciation.

Reviewer 2 Report

Manuscript describing response to airborne ozone and soil borne metal pollution in afforestation plants is very interesting and it deserves to be published, however after some minor changes. 

In Introduction part of the manuscript authors described the aim of the study and hypothesis, but they could describe it in more details. Especially they could focus more on each stress agent and types of responses. In the last part, last paragraphs of the Introduction authors should not describe again what they did, rather focus on the main goal of the presented study.

Question that is arising is why did authors decide to focus on these selected plant species?

Results are presented clearly, in a coherent manner and the manuscript is easy to follow. 

In conclusion authors could present some more future suggestions in this area of research.

Author Response

REVIEWER #2

“Manuscript describing response to airborne ozone and soil borne metal pollution in afforestation plants is very interesting and it deserves to be published, however after some minor changes.”

OUR ANSWER: Thank you very much for the positive appreciation.

“In Introduction part of the manuscript authors described the aim of the study and hypothesis, but they could describe it in more details. Especially they could focus more on each stress agent and types of responses. In the last part, last paragraphs of the Introduction authors should not describe again what they did, rather focus on the main goal of the presented study.”

OUR ANSWER: The current science on penetration routes, effects on plant physiology and triggered injury by ozone and metal stress agents is briefly synthesized in the 2nd and 3rd paragraph of Introduction, respectively, based on the relevant and up-to-date literature. The effects on mineral nutrition of both stress agents are reported in the 4th paragraph. The statements describing our study rationale, objectives, hypotheses and experimental design are grouped by the end of Introduction, in the final paragraphs. Our report plan is thus quite standard. In our view, our statements represent a suited trade-off between extensiveness and conciseness of required reports. In such a study on multiple stress effects and by contrast with preceding work on the studied stress agents taken alone, the main working hypotheses tested are about combined stress effects. Hence our opinion is that the concerns raised by this reviewer are already sufficiently addressed in the original submission.

“Question that is arising is why did authors decide to focus on these selected plant species?”

OUR ANSWER: The justification for the selected model species is provided at the beginning of the study rationale description in the Introduction (i.e. “(…), we used a multifactorial experimental design to compare the responses to different environmental stressors of species representative of the main plant life forms that spontaneously reclaim typical brownfield sites in Central Europe [33]”), with more details provided in the last paragraph of chapter 4.1, from the Material and Methods section. Our study focus was the responses to combined stress agents in afforested trees or plant species spontaneously reclaiming metal-contaminated sites from Central Europe.

“Results are presented clearly, in a coherent manner and the manuscript is easy to follow. In conclusion authors could present some more future suggestions in this area of research.”

OUR ANSWER: Thank you for the positive appreciation. Indeed, one can suggest other research directions beside the mentioned stress-enhancing versus stress-hardening by memory effects after transient stress. One main research gap is certainly the effects of higher temperatures and/or reduced precipitation in combination with metal stress, given the rapidly changing climate worldwide. However, given remarks on the “Conclusion extensiveness” by another reviewer, we have not introduced further research recommendations, considering the length of Conclusion and overall manuscript.

Reviewer 3 Report

Reviewer

MDPI – Plants

Manuscript Number: plants-2540800

Title: « Responses to airborne ozone and soilborne metal pollution in afforestation plants with different life forms »

The article is devoted to the study of the effect of ozone and heavy metals on plants with independent and joint exposure. The topic is very interesting and relevant. However, there are several questions:

Why were these particular types of plants chosen? Why not study the effects on cereals, vegetables and other crops?

What was the hypothesis in studying the combined effects of ozonation and heavy metal pollution?

By what principle was the ozone dose equal to 35 ppm/h chosen?

The conclusion section is very extensive. It should be made more concise.

Author Response

REVIEWER #3

“The article is devoted to the study of the effect of ozone and heavy metals on plants with independent and joint exposure. The topic is very interesting and relevant.”

OUR ANSWER: Thank you very much for the positive appreciation.

“However, there are several questions:

Why were these particular types of plants chosen? Why not study the effects on cereals, vegetables and other crops?”

OUR ANSWER: except in a few cases and for particular diet requirements (e.g. selenium-enriched food), food crops cannot be raised on metal-contaminated sites for food safety reason and the contaminated arable land is then excluded from the economic cycle (see the European critical limits cited in the paper). For making these sites productive again, e.g. fiber, lumber or bio-energy crops, as the two tree species in our experiment, may provide alternatives, if they can stand soil contamination and the processed products are reasonably free of contamination. At other i.e. urbanized sites, crop production is often not an option. One needs then to know how safe the reclaiming vegetation - often including ruderals and pioneer trees - is and which species/genotypes should be promoted for avoiding lateral and vertical contamination spread. The justification for the model species selected in our study is provided at the beginning of the description of the study rationale in the Introduction and further details are provided in the last paragraph of chapter 4.1, from the Material and Methods section.

“What was the hypothesis in studying the combined effects of ozonation and heavy metal pollution?”

OUR ANSWER: combined ozone and heavy metal pollution is an environmental reality at many places worldwide nowadays [see manuscript line 60: “ European pollution maps indicate the coincidence of serious airborne ozone pollution and soilborne metal contamination throughout industrialised regions [6,19]”]. However, current mechanistic understanding on combined abiotic stress effects is still scarce, particularly in the case of airborne ozone and soilborne heavy metal pollution, which represents an important research gap. Two groups of study hypotheses are presented by the end of Introduction section. The first group is about the penetration routes of both stress agents and consequences on the stress effects and interactions. The second group of hypotheses is about the plant responses to the stress factors, their severity and variation according to plant species and organ.

“By what principle was the ozone dose equal to 35 ppm/h chosen?”

OUR ANSWER: we did not select a given ozone concentration or targeted any ozone dose (concentration-based) but reproduced in controlled conditions the ozone pollution and other environmental conditions found at a Swiss ozone hotspot [cf. manuscript line 146: “(…), replicating the environmental conditions at a Swiss ozone hotspot” and line 668-673: “(…) and close replication of environmental and ozone conditions measured from 14 April to 30 September (1998) at an ozone pollution hotspot in Switzerland (Morbio Superiore, N 45°51’41’’/E 9°01’47’’, 599 m a.s.l.; Fig. S1). The temperature and humidity were adjusted every 15 min to correspond to the values at Morbio Superiore during the reference period. The daylight course was simulated, by adjusting the light intensity of 25 lamps per chamber every hour”]. The achieved ozone exposure is representative of values commonly measured in the Mediterranean basin.

“The conclusion section is very extensive. It should be made more concise.”

OUR ANSWER: We would like it! Since it mostly consists in a discussion of the hypotheses (from 2nd statement and beside the final take-home messages in the last three statements), we hardly see how to be more concise without losing important conclusions. However, we have refrained to indicate other important research directions beside the mentioned stress-enhancing versus stress-hardening by memory effects after transient stress, as requested by another reviewer.